# Evaluating the Assembly Strategy of a Fungal Genome from Metagenomic Data: *Solorina crocea* (Peltigerales, Ascomycota) as a Case Study

**DOI:** 10.3390/jof11080596

**Published:** 2025-08-15

**Authors:** Ana García-Muñoz, Raquel Pino-Bodas

**Affiliations:** 1Area of Biodiversity and Conservation, Department of Biology and Geology, Physics and Inorganic Chemistry, University Rey Juan Carlos, Móstoles, 28933 Madrid, Spain; 2Global Change Research Institute (IICG-URJC), University Rey Juan Carlos, Móstoles, 28933 Madrid, Spain

**Keywords:** *de novo* genome assembly, lichenized fungi, metagenomics, functional annotation

## Abstract

The advent of next-generation sequencing technologies has given rise to considerably diverse techniques. However, integrating data from these technologies to generate high-quality genomes remains challenging, particularly when starting from metagenomic data. To provide further insight into this process, the genome of the lichenized fungus *Solorina crocea* was sequenced using DNA extracted from the thallus, which contains the genome of the mycobiont, along with those of the photobionts (a green alga and a cyanobacterium), and other associated microorganisms. Three different strategies were assessed for the assembly of a *de novo* genome, employing data obtained from Illumina and PacBio HiFi technologies: (1) hybrid assembly based on metagenomic data; (2) assembly based on metagenomic long reads and scaffolded with filtered mycobiont long and short reads; (3) hybrid assembly based on filtered mycobiont short and long reads. Assemblies were compared according to contiguity and completeness criteria. Strategy 2 achieved the most continuous and complete genome, with a size of 55.5 Mb, an N50 of 148.5 kb, and 519 scaffolds. Genome annotation and functional prediction were performed, including identification of secondary metabolite biosynthetic gene clusters. Genome annotation predicted 6151 genes, revealing a high number of genes associated with transport, carbohydrate metabolism, and stress response.

## 1. Introduction

Lichens are complex symbiotic associations between a fungus (the mycobiont) and a green alga or cyanobacterium (the photobiont) [1,2]. Recent research has revealed that lichen thalli harbor a high biodiversity of microorganisms, including bacteria and other fungi [3,4,5,6,7,8]. Several photobionts have also been detected in a single lichen thallus [9]. This presents significant challenges in generating *de novo* whole genomes from mycobionts. The isolation and axenic culture of mycobionts is a difficult and frequently unsuccessful task [10,11], which has led to sequencing genomes primarily using metagenomic techniques [12,13,14,15,16,17,18,19]. However, the extraction of the mycobiont genome from a metagenomic assembly is not straightforward and depends on databases available for the taxonomic assignment [13]. Although there is a wide variety of bioinformatic programs and pipelines for this purpose, their accuracy strongly relies on the quality of the starting metagenomes [20]. For instance, MetaWRAP [21] and EasyMetagenome [22] are user-friendly pipelines developed to isolate and annotate genomes from metagenome short read data.

Third-generation sequencing platforms such as Pacific Biosciences (PacBio) and Oxford Nanopore Technologies (ONT) have been demonstrated to increase the continuity of genomes assembled by several hundred-fold [23,24]. Despite the existence of numerous genomes from lichenized fungi, only a limited number of these have been generated using long read sequencing [15,18,25,26,27]. Consequently, the number of chromosomal-level assemblies for these fungi remains limited. However, when data are derived from metagenomics, where diverse organisms are present in uneven proportions, sequencing long reads can also lead to highly fragmented genomes [28]. Consequently, the integration of long and short read sequencing technologies emerges as a promising approach for generating high-quality genomes from metagenomic data. This is because it combines the advantages of short reads, which provide high base accuracy, and long reads, which provide high genome connectivity [29,30]. In contrast, producing similar results using only long reads requires deep sequencing, which leads to an increase in the cost of genomes [31].

There are two main strategies for carrying out the assembly of a genome by combining short and long reads. One is to use short and long reads simultaneously in a hybrid assembly. The second one performs an assembly using the long reads and uses the short reads to polish the generated assembly. Comparisons between these two assembly strategies have been made in bacteria with data from organisms isolated in culture [32,33]. It is difficult to extrapolate and predict which pipeline outperforms others in terms of quality of genomes between different taxonomic groups because it largely depends on the data structure and target genome [34]. A few genomes of lichenized fungi were yielded using both sequencing technologies [15,35], and they used different assembly strategies. Therefore, we are unable to ascertain which of the two strategies is able to generate a higher-quality lichenized fungal genome. The assessment of different assembly methods for generating high-quality genomes of lichenized fungi from metagenomic data can serve as a framework for guiding future research initiatives.

The order Peltigerales encompasses approximately 1300 species [36], including some macrolichens characterized by their notably large thalli. Many of the species of this order have been associated with mature, well-preserved forests with stable environmental conditions. These lichenized fungi establish symbiosis with cyanobacteria, bipartite lichens, or have a green alga as the primary photobiont and a cyanobacterium as secondary photobiont, or tripartite lichens [37]. In the latter case, the generation of high-quality mycobiont genomes from metagenomic data may present an even harder challenge than in bipartite lichens.

In this study, a comparative analysis is conducted on three assembly strategies for the combination of long and short reads from the metagenomics of *Solorina crocea* (L.) Ach., a tripartite lichen symbiosis. The three strategies are: (a) hybrid assembly using long and short reads; (b) assembly based on long reads, using short reads for scaffolding and polishing the genome; and (c) hybrid assembly using reads from the mycobiont, after a filtration process. The prediction of repetitive elements and putative genes, as well as their functional annotation, was conducted in the final assembly, providing new genetic resources of this species for future research.

## 2. Materials and Methods

### 2.1. Characterization of Solorina crocea

*Solorina crocea* is a terricolous species characterized by a foliose thallus with an orange undersurface associated with the green algae *Coccomyxa* and with the cyanobacterium *Nostoc* [38]. It inhabits snow-beds and is widely distributed across arctic–alpine regions of Eurasia and North America [39,40,41,42]. It is a rare species on the Iberian Peninsula, although the populations consist of numerous individuals [43].

### 2.2. Fungal Material

A few specimens were collected in the Puerto de la Quesera, Segovia (41°12′44″ N, 3°25′10″ W), which is located in the Ayllón massif, belonging to the Sistema Central, at an altitude of above 1700 m. The vegetation is composed of *Erica australis* L. heathland, and the higher areas are dominated by grassland on an acidic substrate. From a climatic perspective, the Ayllón massif is characterized by being the wettest in the Sistema Central, influenced by the Golfo de Vizcaya and the Northern Iberian System, with a considerable increase in summer rainfall [44]. Consequently, this area presents a high diversity of plant and lichenized fungi species with a boreal and boreo-alpine distribution [45,46,47]. The specimens studied have been deposited in the MACB herbarium (MACB 132680).

### 2.3. DNA Extraction and Quantification

Specimens were carefully cleaned to ensure the complete removal of all residual substrate. Prior to the DNA extraction, the lichen fragments were immersed in acetone for two hours to remove the secondary metabolites. Subsequently, the acetone extracts were used to identify the secondary metabolites by thin-layer chromatography (TLC), according to the standard procedure described by [48], with two solvents A (180 mL toluene–45 mL dioxan–5 mL acetic acid) and C (170 mL toluene–30 mL acetic acid). The fragments were ground using 3 mm glass beads with a Precellys^®^ 24 (Bertin Technologie, Berlin, Germany) tissue homogenizer. DNA extractions were carried out using the 2% cetyltrimethyl ammonium bromide (CTAB) method [49,50], and the DNA was resuspended in Tris-Cl (10 mM). The DNA extractions were quantified using a Quantus^TM^ Fluorimeter (Promega, Madison, WI, USA), yielding a concentration ≥ 45 ng/µL of DNA for all extractions. The ITS rDNA region was amplified for all extractions using the protocol described in [51], and nucleotide BLAST searches v2.15.0 [52] were conducted to remove contaminations.

### 2.4. Library Preparation and Sequencing

Libraries and sequencing were carried out at Macrogen (Macrogen Inc., Daejeon, South Korea, www.macrogen.com). In order to generate the whole genome of the target species, a combination of short (Illumina Technology, San Diego, CA, USA) and long reads (PacBio Technology, Menlo Park, CA, USA) sequencing was carried out. The TruSeq Nano DNA Kit (Illumina, Inc., San Diego, CA, USA) was used to prepare the short read library and sequenced with 150 bp paired-end reads in NovaSeq 6000. Sequencing was designed to obtain 80× coverage using short reads and 10× coverage using long reads.

### 2.5. Quality Control and Reads Filter

Prior to assembly of the Illumina reads, the adapters and low-quality regions of Illumina reads were removed with Trimmomatic v0.39 [53]. The specific settings established the quality threshold of sliding windows of four bases with a minimum quality of 15, leading and trailing bases with a quality below 3 were removed, and reads with a length below 36 bp were discarded. In the case of long reads, no quality filtering or error correction step was performed prior to assembly.

The universal fungal barcode ITS sequence (ITS1 and ITS2) was extracted using a Python script (extractITS.py; https://github.com/fantin-mesny, accessed on 29 June 2025) based on ITSx v1.1.3 software [54] to confirm the identity of the sequences by a BLASTn search [55].

### 2.6. Benchmarking of Strategies for Genome Assembly

#### 2.6.1. Assembly Strategy 1: Hybrid Metagenome Assembly Using Short and Long Reads

Illumina and PacBio HiFi reads were used simultaneously as input for running a hybrid assembly with metaSPAdes v4.0.0 [56] using the default settings (Figure 1).

A taxonomic assignment of the metagenome contigs was made as a previous step for filtering mycobiont contigs. This was conducted using DIAMOND [57], using Uniref90 [58] as a database and BLAST v2.15.0 searches (e-value cutoff = 1 × 10^−25^) using a customized database. To elaborate on the latter database, all Peltigerales genomes available at JGI and NCBI were inspected using GC plots, which indicate whether the genome may be chimeric, including contigs from other organisms. In total, the database includes 13 genomes of Lecanoromycetes, five of which belong to Peltigerales (Appendix A). The mycobiont contigs were retained using BlobTools v.1.1.1 software [59]. It consisted of the clustering of contigs according to taxonomy assignment, GC content, and coverage. As inputs, we use a file containing read coverage data generated using Minimap2 v.2.28-r1209 [60] and two taxonomic assignments files coming from DIAMOND v2.1.10 and BLAST v2.15.0 searches. The first round of BlobTools generated a summary of read coverage, GC content, and taxonomic identification at the phylum level of all metagenomic contigs. Contigs that matched “Ascomycota” or “no-hit” were retained using the seqkit v2.9.0 function [61]. Contigs with unusually high or low GC content were eliminated as they are likely to be contaminants that have been incorrectly assigned to Ascomycota. To ensure the removal of any remaining contaminants, a second round of BlobTools was implemented using the filtered contigs from the first run as input.

#### 2.6.2. Assembly Strategy 2: Genome Assembly Using Long Reads and Scaffolding Using Short Reads

Three different assemblers, which support metagenomic data, were used to assemble PacBio HiFi long reads: Canu v2.2 [62], metaFlye v2.9.5-b1801 [63], and Hifiasm-meta v0.3-r074 [64]. Hifiasm-meta was developed exclusively for metagenomic data from PacBio HiFi reads [64]. The other assemblers support several sequencing technologies, and a specific configuration of settings for the management of PacBio HiFi reads was selected. In addition, the assemblies from metagenomes previously generated were merged in pairs using the Quickmerge software [65] (Figure 1).

The quality of each metagenome resulting from different assemblers (Canu, metaFlye, and Hifiasm-meta) and their combinations by Quickmerge (Canu + Hifiasm-meta, Canu + metaFlye, and Hifiasm-meta + metaFlye) was evaluated by comparing the number of contigs > 500 base pairs, the length of the largest contig, and the N50 contig. These metrics were calculated in QUAST v5.0.2 [66]. In the assessment, the genome size was not considered because the fungal genome size is overestimated in the metagenome. Additionally, BUSCO (Benchmarking Universal Single-Copy Orthologs; [67]), with the Ascomycota_odb10 database, was used to assess the completeness of metagenomes. Regarding the N50 and BUSCO values, the metagenome resulting from the merging of Hifiasm-meta and metaFlye assemblies achieved the best quality and was selected for the subsequent filter steps that generated the mycobiont assembly (Table 1).

The mycobiont contigs were retained from the selected metagenome by combining two approaches. The first approach was based on running two rounds of BlobTools, following the steps outlined in Strategy 1. The other approach to filtering mycobiont contigs from the metagenome was based on BUSCO v5.8.2. The metagenome was filtered using the complete sequences of BUSCOs to ensure that Ascomycota core genes were retained in the filtered contigs. The contigs obtained with both approaches were combined into a unique file. Duplicated contigs were subsequently removed with a seqkit command. The mycobiont contigs were scaffolded with short reads to obtain a final mycobiont assembly.

#### 2.6.3. Assembly Strategy 3: Hybrid Assembly Based on Mycobiont Reads Previously Filtered

Long and short reads belonging to the mycobiont were used to generate a new hybrid assembly using SPAdes v4.0.0 [68,69]. Mycobiont long reads were extracted from the raw PacBio HiFi reads by mapping them to filtered contigs (obtained with BlobTools and BUSCO as detailed in the previous section from Strategy 2). Mapping was performed using Minimap2.

To extract the mycobiont short reads, a metagenome was first generated by assembling Illumina short reads. We tested the performance of two different assemblers, one of them was MEGAHIT v2.19 [70], with default k-mer (33, 55, 77, and 99), and the other was metaSPAdes, also with default k-mer (33, 55, 77, 99, and 127) settings. According to the contiguity principle, the best short read metagenome assembly was obtained with MEGAHIT. Both approaches, BlobTools and BUSCO, were used to obtain the mycobiont contigs from the long read assembly, were used to filter mycobiont contigs from the MEGAHIT metagenome assembly. A file containing read coverage data was generated using BBMap v39.15 [71]. Additionally, CONCOCT [72], a metagenome binning tool, was used to cut the contigs into 10 kb sections. The binned contigs were combined with the BlobTools results and visualized with a script modified from [19] in R v.4.2.3 [73]. The script was based on the ggplot2 v.3.5.1 package [74]. Bins that matched Ascomycota with similar values for GC content and coverage were combined to generate a metagenome-assembled genome (MAG). The MAG was subjected to a second round of BlobTools to remove contaminants. Last, the results were filtered according to the criteria mentioned above by seqkit commands to extract the mycobiont contigs. These contigs were used to extract the short reads corresponding to the mycobiont by mapping the metagenomic Illumina reads against them with BBMap. These mycobiont short reads were then used in the hybrid assembly along with the mycobiont long reads in this strategy and to scaffold the assemblies generated in Strategy 2, as described below.

### 2.7. Scaffolding and Polishing Mycobiont Assemblies

Scaffolding and gap-closing of the mycobiont assemblies obtained with Strategies 1, 2, and 3 were performed in Redundans v2.0.1 [75]. The mycobiont short and long reads were used for genome polishing. To improve the scaffolded assemblies, polishing tools were used iteratively in subsequent rounds. First, the filtered long reads were mapped to the final assembly using Minimap2, generating the input for Racon v1.5.0 [76]. This step was repeated to run three rounds of Racon. The assembly resulting from Racon was additionally polished using the mycobiont short reads. They were mapped to the Racon assembly using BWA mem v0.7.18-r1243 [77] to generate the required file for Pilon v1.24 [78]. Mapping was iteratively performed for running three rounds of Pilon. Finally, the final mycobiont assembly was obtained.

### 2.8. Quality Assessment of Mycobiont Assemblies

An evaluation of the mycobiont assemblies obtained by means of different strategies was conducted using QUAST v5.0.2 [66]. This assessment was undertaken according to the same criteria as were used for the metagenome selection. Additionally, the genome completeness was assessed using (BUSCO) [79] against “Ascomycota_odb10”, which contains a set of 1706 Ascomycota core genes. The tool BUSCO was run in genome mode and with MetaEuk settings [80] as it is faster for fungi [81].

### 2.9. Repetitive Element Library Construction

Genome assemblies generated with the three strategies were employed to create a customized repetitive element library. Both *de novo* and structure-based approaches were used for the identification of transposons and other repetitive elements. First, repetitive elements were identified *de novo* using RepeatModeler2 v2.0.3 [82]. Then, the structure-based program MITE Tracker [83] was used to identify Miniature Inverted-repeat Transposable Elements (MITEs). The identification of other structure-based repetitive elements was achieved using several programs implemented in the Extensive *de novo* TE Annotator (EDTA) package v2.2.2 [84], such as HelitronScanner [85] for helitrons, TIR Learner v3 [86] for terminal inverted repeat (TIR) transposons, LTRHarvest [87], and LTR_FINDER [88] for long-terminal (LTR) elements.

We removed potential non-transposon protein-coding genes from this repetitive element library. For this purpose, a transposon-free protein database was built. First, the available proteomes of Peltigerales species (Appendix A: *Lobaria pulmonaria*, *Lobaria immixta, Pseudocyphellaria aurata*, *Sticta canariensis*, and *Peltigera leucophlebia*) were combined with the curated Uniprot–Swissprot database. A Blastp search of the resulting protein database against the RepeatPeps library, implemented in RepeatMasker, was performed to remove the transposons. Then, the obtained transposon-free protein database was used to filter the non-transposon protein-coding genes from the repetitive element library by running Blastp. Finally, redundancy was reduced with CD-HIT-EST [89] by clustering the sequences with a 95% similarity threshold, obtaining the final repetitive element library. The sequences in this library were used to determine the proportion of the genome covered by repetitive elements using RepeatMasker [90,91]. This program provides the relative proportion of the two types of transposable elements (Class I or retroelements and Class II or DNA transposons) as well as other types of repetitive elements.

The landscape of divergence between repetitive elements and their consensus sequences was calculated by parsing the results from RepeatMasker by a perl script (https://github.com/4ureliek/Parsing-RepeatMasker-Outputs, accessed on 29 June 2025). This analysis estimates the CpG-corrected Kimura two-parameter distance between each repetitive element and its consensus sequence. To obtain the consensus sequence of each repetitive element, their sequences were aligned and trimmed with MUSCLE [92] and TrimAl [93], respectively. The conserved regions were extracted using Gblocks v0.91b [94], and the final consensus sequences were created with the tool ‘em_cons’ v6.6.0.0 in EMBOSS [95].

In order to compare the abundance of repetitive elements with phylogenetically related species, the same pipeline was employed to annotate the five genomes of the order Peltigerales available in NCBI and JGI. A repetitive element library was constructed for each species, which was used to calculate the proportion of their genomes that were covered by repetitive elements and to plot the landscape of repetitive elements. The completeness of these genomes was also assessed by BUSCO.

### 2.10. Gene Prediction and Functional Annotation

Gene prediction was performed by running four rounds in MAKER2 v3.01.04 [96]. The transcriptome of *Lobaria pulmonaria* (BioProject: PRJNA403314), a closely related species belonging to the same family, was used to provide RNA evidence. As protein evidence, the TE-free protein database of Peltigerales was used (see Section 2.9).

The repetitive regions of the genome were soft-masked using RepeatMasker with the customized TE database and the TE library implemented in RepeatMasker. Transcript and protein evidence were aligned to the genome using Blastx and tBlastx, respectively. Alignments were polished using the Exonerate v2.4.0 software [97] implemented in the MAKER2 pipeline. Only contigs longer than 10,000 bp were considered for annotation [98]. Ab initio gene prediction was achieved using SNAP 2006-07-28 [99] and AUGUSTUS v3.5.0 [100] pre-trained with *Saccharomyces* species parameters. The gene models predicted from each round were filtered to retain those with a maximum AED score of 0.25 and a minimum length of 50 bp for training SNAP in the subsequent rounds. The fourth round of MAKER2 generated the final set of predicted gene models.

The functional annotation of the predicted genes was conducted using a range of databases implemented in the FunAnnotate v1.8.17 pipeline [101]. The Pfam domains and Gene Ontology (GO) terms were detected by InterProScan v5.73-104.0 [102], while proteases were identified by MEROPS [103] and carbohydrate-active enzymes or CAZymes by dbCAN [104]. Additional annotations were added using eggNOG-mapper (emapper v.2.1.12) [105] based on eggNOG orthology data [106] and DIAMOND for sequence searches. More detailed functional descriptions of genes were obtained by BLAST searches against the UniProtKB database. Secondary metabolite gene clusters were predicted using antiSMASH v7.1.0 [107].

The number of genes per scaffold was extracted from the annotation file. The GC content and length of each scaffold were calculated using seqkit. Gene density was calculated by dividing the number of genes on each scaffold by its length. Gene densities were plotted in R using the ggplot2 v.3.5.1 package [74].

## 3. Results

Illumina and PacBio HiFi sequencing generated a total of 27,029,432 and 71,378 reads, respectively. Illumina reads yielded consistent 151 bp paired-end short reads, while PacBio HiFi generated reads with an average length of 5134 bp. Assembly strategies comprised both types of reads, generating genomes with contrasting statistics.

### 3.1. Mycobiont Assembly Resulting from Strategy 1

The metagenome assembly generated with Strategy 1 was characterized by a high degree of fragmentation, with a high number of contigs (Table 1). However, it presents a high N50 value (Table 1). The mycobiont assembly resulting from this strategy comprised 4157 scaffolds > 500 bp, with an N50 value of 63.23 kb (Table 2). The majority of the scaffolds were found to be shorter than 100 kb, with the shortest one being 0.7 kb (Appendix A) and gene densities below 200 genes/Mb (Appendix A). The largest scaffolds contained a mean of 36.3 genes (Appendix A). The largest scaffold was 347.7 kb, with a GC content of 36.1% (Appendix A) and a gene density of 106 genes/Mb. BUSCO analysis indicated a completeness of 95.4% for this assembly, corresponding to the presence of 1627 (94%) complete single-copy orthologs.

The annotation predicted a total of 7352 genes (Table 3), with >50% of genes predicted to be well-supported (AED score < 0.1). The BUSCO analysis yielded an annotation completeness estimate of 89.4%. Functional annotation assigned 30,404 functional terms. The total number of genes with at least an annotated function was 6411 (87.2%). A total of 19 biosynthetic gene clusters (BGCs) were identified (Table 3). A total of 10,136 repetitive elements were identified, covering 22.53% of the mycobiont genome. Retrotransposons and DNA transposons were present at similar frequencies, covering 11.34% and 10.26% of the genome, respectively (Appendix A).

### 3.2. Mycobiont Assembly Resulting from Strategy 2

All metagenome assemblies using exclusively long reads showed similar values of GC content and length of the largest contig. The assembly produced by metaFlye exhibited the greatest fragmentation, with an elevated number of contigs. The assemblies generated by merging the assemblies produced by Canu + metaFlye or Hifiasm-meta + metaFlye exhibited reduced fragmentation and enhanced continuity. Despite the Canu + metaFlye assembly exhibiting a lower number of contigs and the largest contig being larger than the Hifiasm-meta + metaFlye assembly, its N50 and BUSCO values were lower, indicating worse genome completeness. Therefore, the Hifiasm-meta + metaFlye assembly was considered the optimal quality for the generation of the mycobiont assembly.

This strategy yielded a mycobiont genome with higher quality than the other strategies in terms of continuity, containing 519 scaffolds > 500 bp and higher values of N50. Most scaffolds were shorter than 200 kb (Appendix A). The largest scaffold comprised 547.6 kb, with a GC content of 35.5% (Appendix A). Most scaffolds presented gene densities greater than 150 genes/Mb (Appendix A), and the largest scaffolds (500 kb–1 Mb) contained a mean value of 50 genes (Appendix A). The completeness analysis, based on BUSCO, estimated a completeness of 96.7%, corresponding with 1633 (95.7%) conserved single-copy orthologs (Table 2).

The annotation predicted a total of 6151 genes (Table 3), with more than 60% of the predicted genes being well-supported. BUSCO analysis estimated an annotation completeness of 91.6%. Functional annotations yielded a total of 52,648 functional terms annotated with different databases (Appendix A). The total number of genes with a predicted function annotation was 5649 (91.8%). AntiSMASH identified 18 biosynthetic gene clusters. A total of 813 repetitive elements were identified, covering 22.18% of the mycobiont genome (Table 3).

### 3.3. Mycobiont Assembly Resulting from Strategy 3

The mycobiont assembly resulting from this strategy was very fragmented, comprising 6280 scaffolds > 500 bp and with the lowest N50 values (Table 2). Most scaffolds were shorter than 50 kb (Appendix A). The largest scaffold has 140.1 kb and a GC content of 42.5% (Appendix A). The gene density was 186 genes/Mb. Most of the scaffolds have a gene density below 200 genes/Mb (Appendix A), while the largest scaffolds contained a mean of 19 genes (Appendix A). This assembly showed the lowest completeness, as indicated by the BUSCO analysis (Table 2).

Gene prediction identified a total of 2767 genes, and more than 60% of these were well-supported. BUSCO analysis estimated an annotation completeness of 33.5%. Functional annotation assigned 14,611 functional terms (Table 3). The total number of genes with at least an annotation function was 2368 (85%). Ten biosynthetic gene clusters were identified using antiSMASH (Table 3).

### 3.4. Functional Features of the Solorina crocea Genome

These features correspond to the annotation of the mycobiont assembly resulting from Strategy 2, which had a higher quality in terms of continuity and completeness (Table 2 and Table 3). The annotation of the assemblies from Strategies 1 and 3 can be found in Appendix A (Appendix A).

The number of DNA transposons identified was greater than that of retroelements. However, retroelements represented a higher percentage of the genome. Specifically, retrotransposons covered 7.65% of the total genome, while DNA transposons covered 5.85%. The most abundant subclass of retrotransposons was the LTR elements, which constituted 7.01% of the whole genome and were classified mainly into Gypsy/DIRS1 and Ty1/Copia. Other non-LTR retrotransposons, such as LINEs, comprised only 0.64% of the retrotransposons, while SINEs were absent. Five superfamilies of DNA transposons were identified: Tc1-IS630-Pogo (3.64%), hobo-Activator (0.41%), MULE-MuDR (0.13%), PiggyBac (0.11%), and Tourist/Harbinger (0.04%). The proportion of repetitive elements that remained unclassified was 7.20%. Simple and low complexity repeats were in a very low proportion (0.62 and 0.18%, respectively) while no satellites were found. The landscape of divergence between the most frequent groups of repetitive elements, including the unclassified elements, showed a low level of divergence (<10%) compared to their consensus sequence (Appendix A). The proportion of repetitive elements found in the genomes of other Peltigerales species was: 18.06% in *Sticta canariensis*, 12.85% in *Pseudocyphellaria aurata*, 31.17% in *Peltigera leucophlebia*, 14.07% in *Lobaria pulmonaria*, and 15.92% in *Lobaria immixta*. Retroelements represented the major fraction of repetitive elements in all genomes (Appendix A). The landscape of repetitive elements also showed a low divergence in all these species (Appendix A).

InterPro annotation, which comprised several databases, retrieved the majority of functional terms, especially from Pfam domains and Gene Ontology (GO). More than half of the genes matched GO terms categorized as molecular function, followed by the biological processes category (Figure 2a). The most frequently functionally annotated terms within these categories were those involved in transport activity, carbohydrate metabolism, and protein activity (Figure 2b). Specifically, the more frequent terms found in the biological processes category were “transmembrane transport”, “translation”, “protein phosphorylation”, and “carbohydrate metabolic process” while the more frequent terms within the molecular function category were termed as “ATP binding”, “protein binding”, “RNA binding”, and “structural constituent of ribosome” (Figure 2b). Pfam annotation mainly yielded protein domains for kinases, transporters, and stress-related genes (Figure 3). The most frequent term was “protein kinase domain”. Specific transport-related domains were noted as “sugar (and other) transporter”, “Major Facilitator Superfamily”, and “Ammonium Transporter Family”, while stress-related domains were noted as “Cytochrome P450” and “TCP-1/cpn60 chaperonin family” (Figure 3). Detailed functional descriptions from UniProtKB complemented the general processes identified by GO terms and Pfam domains. These descriptions involved stress responses, activation of signal transduction, and protein folding and stability (Figure 4). In particular, the most frequent terms were noted as “T-complex protein 1 subunit epsilon”, “Ribosome-associated molecular chaperone SSB1”, “Peroxisomal hydratase-dehydrogenase-epimerase”, “Glycogen synthase kinase 1”, “ER-derived vesicles protein ERV14”, “Elongation factor 3”, “Chitin synthase export chaperone”, and ATP-dependent DNA helicase II subunit 2” (Figure 4). Similar terms as those found in the other annotations were also identified by a homology search; for example, “Cytochrome P450 monooxygenase ORF9”, “Ammonium transporter 1”, and “GTP-binding protein ypt1” (Figure 4).

The genome analysis in antiSMASH of *S. crocea* revealed 18 regions containing biosynthetic gene clusters (BGCs): 5 terpene synthases, 4 post-translationally modified peptides-like (fungal-RiPP-like), 2 polyketide synthases type I (T1PKS), 1 polyketide synthases type III (T3PKS), 2 non-ribosomal peptide synthetases (NRPS), 1 NRP-metallophore/NRPS, 1 NRPS/indole, and 2 NRPS/T1PKS. The detected 18 regions cover 784 kb, comprising only 1.4% of the genome.

## 4. Discussion

The assembly of genomes from metagenomic data is a considerably more complex process than conventional assembly, which starts from an isolated genome. The presence of species at different abundances and species phylogenetically related within the sample that share genomic regions represent the most challenging factors during the process of genome assembly from metagenomic data [28,108]. Nevertheless, metagenomic techniques have become essential for the sequencing of genomes of certain organisms, including lichenized fungi. In this study, the *de novo* genome of *Solorina crocea* was generated from metagenomic data that were sequenced using PacBio and Illumina technologies. This assembly was compared with those generated in previous studies of other Peltigerales species (see Appendix A). Based on continuity and completeness criteria, this is currently the best quality genome reported for the order.

### 4.1. Assessing Different Assembly Strategies and Other Considerations

It is postulated that genomes assembled using multiple sequencing techniques generate better quality assemblies than those assembled using only short or long reads [109,110]. In accordance with the strategies employed in preceding studies for genome assembly combining short and long reads [30,109,111], three strategies were utilized in the present study. Our results clearly demonstrate that the assembly method has a great influence on its quality. In this study, starting from the same dataset, very different results were obtained, in terms of continuity and completeness, depending on the strategy and assembler used (Table 1, Table 2 and Table 3). The findings of this study indicated that the most effective strategy was Strategy 2, which generated an assembly based on long reads, followed by improvement through scaffolding and polishing processes with the filtered mycobiont reads. Despite that, hybrid strategies have been shown to produce high-quality assemblies in other studies, even using metagenomic data [111]. The results obtained from our analyses indicated that both Strategy 1 and Strategy 3 resulted in highly fragmented assemblies, in contrast to Strategy 2. One plausible explanation for the fragmentation of the genome resulting from Strategy 1 is the retention of numerous short contigs. Indeed, the total number of scaffolds contained within this assembly was 10,136, most of them < 500 bp. Nevertheless, the final assembly of Strategy 1 showed higher quality metrics than the assembly generated by Strategy 3, as indicated by the number of scaffolds, N50, and BUSCO scores (Table 2). Regarding annotation, Strategies 1 and 2 appear to be similar. Indeed, the number of predicted genes is higher than that of Strategy 2 (Table 3). The most substantial discrepancy was observed in the number of repetitive elements identified, which was higher in the more fragmented assemblies. The hybrid strategy was previously employed to assemble the genome of the mycobiont *Lasallia pustulata* from metagenomic data [35]. The assembly obtained was of a notably high quality, with 49 scaffolds and an N50 = 1.8 Mb. The observed discrepancies with our results may be attributable to the depth of long read sequencing. As demonstrated in [35], the sequencing coverage was critical for the quality of the final mycobiont assembly. Other studies have also proved the dependence of the number of scaffolds on the coverage of long reads sequenced by PacBio HiFi [20]. Nevertheless, sequencing using PacBio technology remains expensive in comparison with the costs of Illumina technology. Therefore, the addition of the Illumina reads in posterior steps, such as scaffolding and polishing, could be a highly effective strategy to improve the quality of the final genome, as implemented in this study. Scaffolding the filtered contigs with other kinds of sequencing data has been demonstrated to yield high-quality genomes in lichenized fungi; for example, the assembly of the *Acarospora socialis* genome resulted from an assembly based also on PacBio reads, which was scaffolded using Omni-C data [25]. In accordance with prior studies [111,112], our results indicate that the final polishing step (implemented here using Pilon) is pivotal in improving genome quality.

The critical step in Strategy 3 was the extraction of mycobiont reads after the binning process. This assembly was performed with 43.5% of the original long reads and 11.26% of the original short reads. Despite these data being derived from metagenomics, the mycobiont is the most abundant organism within the lichen thallus, and the recovery of a greater number of reads from it was to be expected. The mapping process is subject to bias associated with the accuracy of aligning short reads in multiple positions [113].

Regarding the assemblers used in Strategy 2, large differences were observed among the assemblies. The marked differences in assembly performance between different long read-based assemblers are to be expected and have been demonstrated by extensive benchmarking works [112,114,115]. They may be explained by their efficiency in managing metagenomic data. For example, metaFlye performs well for non-uniform read coverage distribution and can assemble the genome of species highly abundant in the sample [63]. In contrast, Hifiasm-meta is based on an algorithm able to recover also low-abundant organisms [116]. Hifiasm-meta has yielded successful results in the assembly of fungal genomes, including some derived from metagenomic data [117,118,119]. Indeed, its suitability to obtain the best genomes and metagenomes from HiFi reads has been demonstrated by [120] in an exhaustive comparison across several datasets. On the other hand, the benchmarking of long reads assemblers by [112] demonstrated that differences in the genome quality achieved by different assemblers are stronger for low-depth data, which could explain the disparity found in this study.

An additional step was implemented in the pipeline by merging two assemblies from long reads. However, the best result was yielded by merging the metagenome assemblies of metaFlye and Hifiasm-meta, resulting in a less fragmented assembly with larger contigs than the original ones and with high values of N50 and BUSCO. The accuracy of these two assemblers working separately was previously demonstrated [112,120]. The combination of assemblies produced by different programs was expected to create more contiguous and accurate assemblies [65]. This strategy was the one used to assemble the genomes of two species belonging to the lichenized genus *Letharia*, by combining the contigs generated with metaFlye and Canu [15].

The availability of enough genomes from phylogenetically related species is critical for the step of metagenome filtering [121]. However, contamination has been detected in numerous published genomes [122]. Indeed, the customized Peltigerales database was considerably reduced because contamination was detected in many of the available genomes, including one from the species target, *S. crocea*. Other assemblies of Peltigerales were excluded from this database because only the metagenomes are published [123], and mycobiont scaffolds are not available.

Despite the assembly generated by Strategy 2 being less fragmented than the assemblies obtained by Strategies 1 and 3, the number of scaffolds is high. This is due to the fact that the average length of reads obtained using PacBio is relatively short (average = 5012 bp). It should be noted that the length of the fragments obtained during the DNA extraction process has a significant impact on the subsequent library and sequencing processes. The combination of HiFi with Hi-C or generated ultra-long reads (ca. 4 Mb) by Nanopore might be more efficacious if the objective is to assemble a genome at the chromosomal level. Nevertheless, these technologies can be costly and are not essential for many genomic applications. In addition, higher coverage sequencing using Illumina and PacBio could improve the *S. crocea* genome. Regarding the quality of the genome annotation, RNA sequencing from this species is needed to improve it.

### 4.2. Genomic Features of Solorina crocea

The *S. crocea* assembly comprises 519 scaffolds and a genome size of 55.50 Mb, which is consistent with the genome size range of other Peltigerales genome assemblies (16.85–130.50 Mb) [17,124], although the genome size of our assembly is notably smaller than another *S. crocea* genome (93.7 Mb) previously assembled. Unfortunately, no flow cytometry data exist for this species that would confirm which is the most accurate value. Our examination of this other assembly of *S. crocea* (Bioproject PRJEB77567) revealed the presence of putative non-fungal scaffolds or other artifacts (Appendix A). This finding could potentially explain the difference in genome size between the two assemblies.

Many of the functional terms found in *S. crocea* are commonly reported in fungal genomes [125,126,127,128,129,130], including major facilitator superfamily (MFS) and aldo-keto reductases, transmembrane transport, cytochrome P450, protein kinases, ATP binding, and chitin synthases. MFS and aldo-keto reductases are relevant in carbon metabolism [131,132]. Chitin synthases are involved in fungal cell wall formation [133], but variations in these genes have been observed between species with different lifestyles [134]. The cytochrome P450 superfamily is implicated in numerous physiological processes, including the biosynthesis of secondary metabolites [135,136]. An expansion of this superfamily has been detected in lichenized fungi belonging to Lecanoromycetes [130]. Protein kinases participate in pathways associated with stress responses in fungi [137]. It has been demonstrated that certain specific kinases are pivotal for the establishment of lichen symbiosis and are conserved across lichenized fungi [138].

Other gene families detected in the genome of *S. crocea* in high abundance have relevant functions in lichenized fungi. The relatively high frequency of ammonium transporters might also be associated with lichenization, as these genes have been shown to be involved during the establishment of mycobiont–photobiont contact [139]. These transporters in *S. crocea* could be involved in the transfer of ammonium from the cyanobacterial partners to the mycobiont [140]. Functional analysis also yielded the term noted as “sugar and other transporters”, which also comprises organic alcohols according to its definition in InterPro. Thus, it can refer to ribitol transporter, expanded in lichenized fungi [130], apart from the transporter for glucose, the carbon source that cyanobacterial photobionts transfer to mycobionts [141].

The repetitive element content in fungal genomes is highly variable, ranging from 0 to 30% [142]. Within the genomes of lichenized fungi, a considerable variation in repetitive elements content has also been observed, ranging from approximately 1% found in *Gyalolechia flavorubescens* to 21.26% found in *Umbilicaria pustulata* [125,130,143,144,145]. The proportion of repetitive elements identified in the genome of *S. crocea* is 22.18%, which falls within the range of lichenized fungi and within the range identified in the genome of the Peltigerales species annotated in this study (from 12.85 to 31.17%). Within this order, the variation in the content of repetitive elements in Lobariaceae genomes was less than in Peltigeraceae. However, the number of repetitive elements identified in *S. crocea* and *P. leucophlebia* was very similar (813 and 829, respectively). Regarding the proportion of each of the types of repetitive elements, the most abundant in *the S. crocea* genome were the LTRs, which have been found to be the most abundant in fungal genomes [142,146]. In other fungal groups, the prevalence of repetitive elements has been associated with a pathogenic lifestyle [147,148]. *Solorina crocea* also presents a high proportion of DNA transposons. These and the other types of repetitive elements found in *S. crocea* present a low divergence with respect to the consensus sequences, which was similarly found in the repetitive elements of *U. pustulata* [145].

Secondary metabolites in lichens can represent a significant proportion of the dry weight of the thallus [149,150,151]. Therefore, it was expected to find a high number of biosynthetic gene clusters (BGCs) in the genome of *S. crocea*. The majority of BGCs identified in the genome of *S. crocea* are associated with terpene synthesis. Numerous biological functions have been attributed to these compounds, including the establishment of symbiotic relationships [152,153]. Consequently, it is logical to predict the presence of a high number of these clusters in the genomes of lichenized fungi [144,154,155,156,157]. The two large terpene cluster groups (Clan 1 and Clan 2) identified by [156] in lichenized fungi have been detected in Peltigerales.

Additionally, BGCs related to polyketide pathways have also been detected. Some of them should be involved in the synthesis of anthraquinones, norsolorinic acid, and averantin, which have been previously reported in *S. crocea* [158]. RiPPs have been studied little in Ascomycota, and their biological function in lichenized fungi is unknown [159]. However, recent research has shown that RiPPs are very common in the genomes of lichenized fungi [159,160]. A total of four cluster RiPPs have been detected in the genome of *S. crocea*, which is within the range found in the genomes of other Peltigerales species, although their abundance is larger in other species from the order than in *S. crocea* [159,160]. NRPS are also widespread among a large variety of Lecanoromycetes species [26,156,159,161]. Three regions encoding PKS are present in the *S. crocea* genome. PKS are multifunctional enzymes involved in the synthesis of polyketides [162]. These compounds constitute the major group of secondary metabolites synthesized by lichenized fungi [162].

## Figures and Tables

**Figure 1 jof-11-00596-f001:**
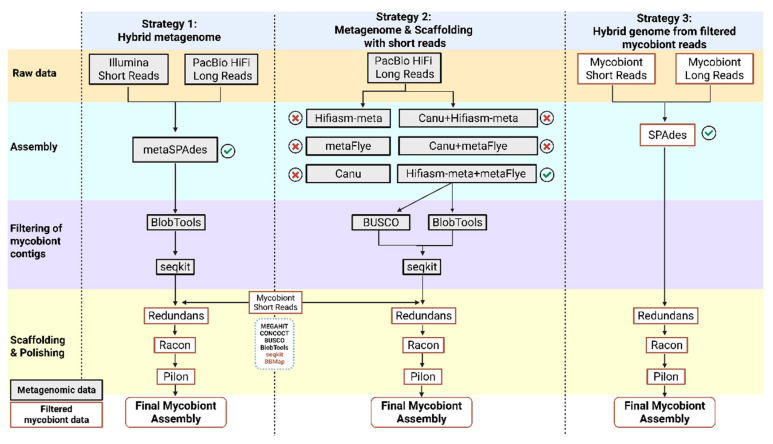
Bioinformatics workflow describing the three strategies tested to assemble the genome of *S. crocea*. The figure indicates the programs used in each step. Metagenomic and filtered mycobiont data are represented in different colors, as shown by the legend. The red cross indicates the assemblies discarded because of worse continuity and completeness values. The green mark indicates the selected assemblies and the assembly used as an input in the following steps. Created in Biorender. Ana García-Muñoz (2025) https://app.biorender.com/citation/6899a743b180816c448a7908.

**Figure 2 jof-11-00596-f002:**
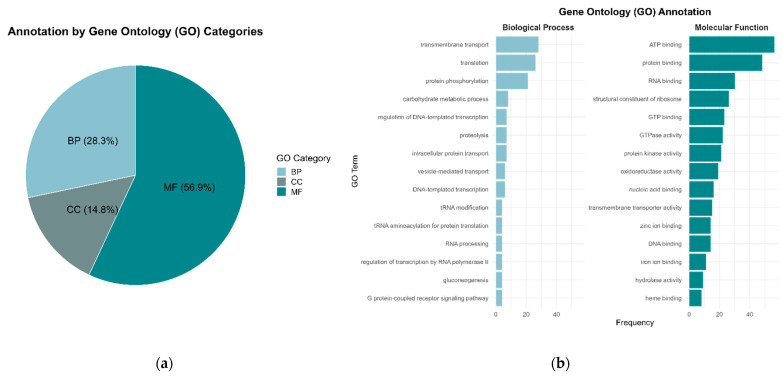
Functional classification of *S. crocea* genes based on gene ontology (GO) annotation. (**a**) Proportion of genes annotated by each GO category (BP: biological processes, MF: molecular function; CC: cellular components); (**b**) Top 15 of more abundant GO terms of the most represented categories.

**Figure 3 jof-11-00596-f003:**
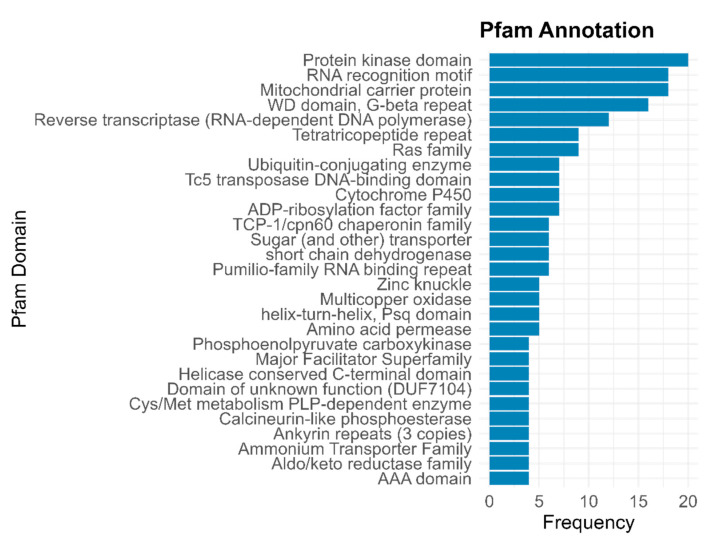
Result of functional annotation carried out by Pfam domains. The most abundant functional terms are shown.

**Figure 4 jof-11-00596-f004:**
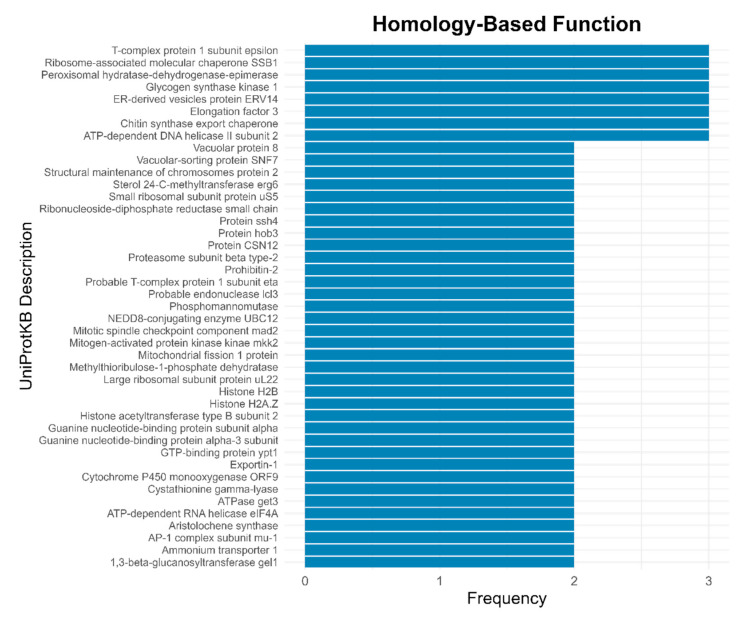
More frequent homology-based functional annotations retrieved from searches against the UniProtKB database.

**Table 1 jof-11-00596-t001:** Comparison of metagenome results using Strategies 1 and 2. The selected metagenome from Strategy 2 for mycobiont contigs filtering and reads extraction is shaded. Assembly metrics were obtained using QUAST. BUSCO scores stand for complete (C), complete and single-copy (S), complete and duplicated (D), fragmented (F), and missing (M) BUSCOs.

Sequencing Technology	Assembly Tool	No. Contigs	Largest Contig (bp)	Genome Size (Mb)	N50 (kb)	%GC Content	%BUSCO Scores
Strategy 1: hybrid assembly							
PacBio HiFi + Illumina	metaSPAdes	22,583	347,633	121.69	41.75	39.16	C: 97.8 [S: 96.2, D: 1.6], F: 0.5, M: 1.7
Strategy 2: long reads + short reads scaffolding							
PacBio HiFi	Canu	4834	242,057	68.16	15.31	41.34	C: 43.0 [S: 42.1, D: 0.9], F: 5.9, M: 51.2
PacBio HiFi	Hifiasm-meta	4804	230,570	108.66	25.57	41.61	C: 67.3 [S: 65.2, D: 2.1], F: 5.3, M: 27.4
PacBio HiFi	metaFlye	6089	230,570	118.01	24.83	41.56	C: 68.6 [S: 66.8, D: 1.9], F: 5.3, M: 26.0
PacBio HiFi	Canu + Hifiasm-meta	4802	327,659	109.03	25.65	41.61	C: 67.4 [S: 65.2, D: 2.1], F: 5.3, M: 27.3
PacBio HiFi	Canu + metaFlye	3941	409,251	96.59	29.29	41.41	C: 60.8 [S: 59.2, D: 1.6], F: 5.8, M: 34.2
PacBio HiFi	Hifiasm-meta + metaFlye	4325	264,302	117.58	32.54	41.68	C: 72.6 [S: 70.3, D: 2.3], F: 4.7, M: 22.6

**Table 2 jof-11-00596-t002:** Final mycobiont assemblies generated by Strategies 1, 2, and 3, after scaffolding and polishing. The table shows the parameters used to assess the continuity and completeness of the assemblies. These metrics were obtained by QUAST. BUSCO scores stand for complete (C), complete and single-copy (S), complete and duplicated (D), and fragmented (F) and missing (M) BUSCOs.

Assembly Strategy	No. Scaffolds	Largest Contig (bp)	Genome Size (Mb)	N50 (kb)	%GC Content	%BUSCO Scores
Strategy 1: hybrid assembly	4157	347,633	90.81	63.23	37.46	C: 95.4 [S: 94.0, D: 1.4], F: 0.6, M: 4.0
Strategy 2: metagenome & scaffolding with short reads	519	547,588	55.50	142.84	37.24	C: 96.7 [S: 95.7, D: 0.9], F: 0.7, M: 2.6
Strategy 3: hybrid assembly of filtered long and short reads	6280	140,100	33.71	16.58	42.43	C: 37.6 [S: 37.5, D: 0.1], F: 1.0, M: 61.4

**Table 3 jof-11-00596-t003:** A comparison of annotations derived from assemblies obtained by means of the three strategies. BUSCO scores stand for complete (C), complete and single-copy (S), complete and duplicated (D), and fragmented (F) and missing (M) BUSCOs.

	Strategy 1	Strategy 2	Strategy 3
No. genes	7352	6151	2767
Genes < 0.1 AED score	56%	61%	53%
BUSCO (annotated gene set)	C: 89.4% [S: 88.1%, D: 1.3%], F: 2.9%, M: 7.7%	C: 91.6% [S: 90.7%, D: 0.8%], F: 2.8%, M: 5.7%	C: 33.5% [S: 33.5%,D: 0%], F: 1.6%, M: 64.8%
No. repetitive elements	10,136	813	8739
Genome covered by repetitive elements	22.53%	22.18%	22.04%
BGCs	19	18	10
Genome covered by BGCs	0.25%	1.40%	0.23%
Functional terms	30,404	52,648	14,611
Annotated genes	87.2%	91.8%	85.0%

## Data Availability

The genome and its structural annotation have been deposited at GenBank under the accession JBODMP000000000 (BioProject: PRJNA1262437) and will be released after publication. Code is currently stored in a GitHub repository (https://github.com/anagarciamu15/Assembly-Strategies, accessed on 29 June 2025) and will be made publicly available upon acceptance of the manuscript.

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
