# Peer review of "Evaluating the Assembly Strategy of a Fungal Genome from Metagenomic Data: Solorina crocea (Peltigerales, Ascomycota) as a Case Study"

_jof, 2025, doi:10.3390/jof11080596_

Round 1
Reviewer 1 Report
The authors present results from a comparative study of methods to assemble fungal genomes from metagenomic data, using the lichen Solorina crocea. This lichen was chosen because it is a tripartite lichen, containing both green algae and cyanobacteria, making the metagenomic data even more complex than bipartitie lichens. Lichens in general present a unique challenge in assembling species genomes of high quality, due to their symbiotic nature and the difficulty in growing axenic cultures of mycobionts in the lab. This paper addresses a relevant and timely challenge, which will be useful for lichenologists and applicable to other intimate symbiotic systems.
The authors are thorough in their methods descriptions, and make difficult concepts accessible through detailed explanations and figures. I only have some minor corrections in the text, see below.
Line 2: In the title, consider changing it to “Searching for the most effective…”
Line 32: Change to “Recent research has revealed…”
Line 44: “have been demonstrated….”
Line 45: change “folds” to “fold”
Line 74: remove “Therefore,”. Start with “In the latter case,...”
Line 87: “foliose” is a more common term than “foliaceous” for lichens, but the authors can decide which sounds best
Line 138: change “assembly” to “assembled”
Figure 1: This is a very informative Figure, but it would be really helpful to make the text boxes larger and therefore easier to read. In addition, if you can, try to reduce the opacity (increase transparency) of the background colors. Also please ensure that these color choices are colorblind friendly if you can.
I think there is s typo under Strategy 2: “Bloodtools” should be “BlobTools”
Line 153: When you talk about the “quality of each metagenome,” are you referring to comparisons between the three strategies, or comparisons between multiple metagenome/sample replicates? Please clarify
Line 229: remove “Finally,” begin with “An evaluation of…”
Lines 247-255 are a bit repetitive. Maybe begin by describing how you built the transposons-free protein database, then describe how you used that to filter non-transposons protein-coding genes from the repetitive element library.
Line 267: I think “MUSCLE” is supposed to be before the citation [90]
Line 317: change “compared to the obtained” to “compared to that obtained”
Line 401: consider changing “different” to “multiple”
Reviewer 2 Report
And I found that this work has organized a workflow to assemble a fungal genome called Solorina crocea from metagenomic datasets, trying to be a general assembling example for other genomes of fungi, this is invaluable. However, the present manuscript could hardly describe the full process to help readers realize its procedures clearly, so I think the authors should consider to polish their manuscript draft to make it understandable. However, there are some issues for improvement in the manuscript.
Major comments:
- The figures need to be improved and share the reproducible script and data in GitHub or Gitee. Such as ImageGP 2 (DOI: 10.1002/imt2.239) can generate high quality figures and with reproducible scripts.
- The structure of the results appears not in concise. The authors should have presented their findings in around 3 mainly sections will be better.
- Figure 1 need to replot by Biorender to reorganized the workflow to more colorful and attractive.
- Recently, published EasyMetagenome is quickly popular for metagenomic data analysis, please compare it and should the advantages of this study.
Minor comments:
- The whole manuscript is full of lengthy words and the sentences structure is far from clear to make the work understandable. For example: (1) In the abstract section: Lines 19 and 20 “long-reads” and “short-reads”, I think use the “long reads” and “short reads” is enough. (2) Lines 22 and 23 “with 55.5 Mb in size, N50 = 148.5 Mb and 519 scaffolds”, should be clearly revised like “with a size of 55.5 Mb, an N50 of 148.5 Mb, and 519 scaffolds”. (3) line 23 “identification of gene clusters of secondary metabolites” could be simply clarified like “identification of gene clusters related to secondary metabolites”. (4) line 25 “high amount of genes related to” could be revised like “high number of genes associated with”, etc. Thus, you can make your manuscript fluency enough to make your readers understand your work easily.
- In the rest of your manuscript, you should follow the suggestion I provided in comment 1 to revise your manuscript. For example: In lines 32-34, “Recent researches have revealed”, I think you should use “Recent research has revealed” instead. In lines 34 “Several photobionts have been also detected in a single lichen thallus.” could simply revised as “ Several photobionts have also been detected”, please always avoid making your sentences uneasy to read.
- In lines 86, the title “Species of study”, I think should rewrite like “ Characterization of the Solorina crocea” instead.
- Please make revision on the procedure you employed in this manuscript to make them easy to understand.
In conclusion, the manuscript might have provided a solution for fungal genomes assembly, but the present manuscript should be carefully revised by the authors further.
Author Response
Major comments
And I found that this work has organized a workflow to assemble a fungal genome called Solorina crocea from metagenomic datasets, trying to be a general assembling example for other genomes of fungi, this is invaluable. However, the present manuscript could hardly describe the full process to help readers realize its procedures clearly, so I think the authors should consider to polish their manuscript draft to make it understandable. However, there are some issues for improvement in the manuscript.
Major comments:
- The figures need to be improved and share the reproducible script and data in GitHub or Gitee. Such as ImageGP 2 (DOI: 10.1002/imt2.239) can generate high quality figures and with reproducible scripts.
ANSWER: Thanks for the comment. We have improved the quality and reproducibility of Figures: 1) The Fig 1 was improved using Biorender, by increasing its legibility; 2) A complete script, including the one used to generate the figures, is available at GitHub. The link is included in the section “Data Availability Statement” of the manuscript.
Regardless, the original figures are of a higher quality than those in the manuscript and will be provided to the journal once the manuscript is accepted for publication.
- The structure of the results appears not in concise. The authors should have presented their findings in around 3 mainly sections will be better.
ANSWER: The results have been divided into four subsections, one for each of the assembly strategies and the fourth presenting the functional characteristics of the Solorina crocea genome.
- Figure 1 need to replot by Biorender to reorganized the workflow to more colorful and attractive.
ANSWER: Thank you for this suggestion. The figure has replot in Biorender a more colorful figure. We re-adjusted its colors to be accessible for color-blind readers.
Recently, published EasyMetagenome is quickly popular for metagenomic data analysis, please compare it and should the advantages of this study.
ANSWER: We agree that EasyMetagenome is a valuable, robust and user-friendly pipeline that offer a complete set of tools for shotgun metagenomic analysis producing reliable results. However, this robust workflow is optimized particularly for prokaryotic organisms from microbiome research sequenced by short-read technologies (Bai et al 2025. Imeta, 4(1), e70001). Currently, this pipeline is not optimised for assembling long reads data from metagenomics and is therefore not suitable for addressing our aim. Nevertheless, given the potential of this tool, reference to it is made in the introduction (lines 40-43).
Detailed comments
Minor comments:
- The whole manuscript is full of lengthy words and the sentences structure is far from clear to make the work understandable. For example: (1) In the abstract section: Lines 19 and 20 “long-reads” and “short-reads”, I think use the “long reads” and “short reads” is enough. (2) Lines 22 and 23 “with 55.5 Mb in size, N50 = 148.5 Mb and 519 scaffolds”, should be clearly revised like “with a size of 55.5 Mb, an N50 of 148.5 Mb, and 519 scaffolds”. (3) line 23 “identification of gene clusters of secondary metabolites” could be simply clarified like “identification of gene clusters related to secondary metabolites”. (4) line 25 “high amount of genes related to” could be revised like “high number of genes associated with”, etc. Thus, you can make your manuscript fluency enough to make your readers understand your work easily.
ANSWER: Thanks. All suggestions proposed by the reviewer has been included. Additionally, the whole manuscript has been revised to make it more readable.
In the rest of your manuscript, you should follow the suggestion I provided in comment 1 to revise your manuscript. For example: In lines 32-34, “Recent researches have revealed”, I think you should use “Recent research has revealed” instead. In lines 34 “Several photobionts have been also detected in a single lichen thallus.” could simply revised as “ Several photobionts have also been detected”, please always avoid making your sentences uneasy to read.
ANSWER: Done. As mentioned above, we followed your suggestions in the rest of the manuscript.
- In lines 86, the title “Species of study”, I think should rewrite like “ Characterization of the Solorina crocea” instead.
ANSWER: Done.
- Please make revision on the procedure you employed in this manuscript to make them easy to understand.
ANSWER: The manuscript has been revised and several sentences corrected and clarified.
In conclusion, the manuscript might have provided a solution for fungal genomes assembly, but the present manuscript should be carefully revised by the authors further.
Reviewer 3 Report
This study fails to deliver on its promise to identify "the most effective assembly strategy" for fungal genomes from metagenomic data. The evaluated strategies—hybrid metagenomic assembly (1), long-read assembly with short-read scaffolding (2), and hybrid assembly of filtered mycobiont reads (3)—yielded universally suboptimal results, with Strategy 2 (55.5 Mb, N50=142.8 kb, 519 scaffolds) declared "best" despite high fragmentation and failure to achieve chromosome-scale continuity. Crucially, Strategy 3 performed catastrophically (N50=16.6 kb, BUSCO=37.6%), yet the authors omitted root-cause analysis for this failure. The conclusions lack rigor due to unaddressed limitations: PacBio read lengths (avg. 5 kb) were insufficient for complex metagenomes, no benchmarking against state-of-the-art tools (e.g., metaFlye+Hi-C) was conducted, and claims of strategy superiority remain unsubstantiated beyond the authors' compromised dataset. The title overstates both methodological innovation and outcomes, as the work merely confirms known challenges without advancing scalable solutions.
PacBio reads averaged 5,012 bp—insufficient for resolving repetitive regions in complex metagenomes.
In fact, I am more in favor of the outcome of Strategy 1, but the author gave up on it because it was too fragmented. The reason why Strategy 1 is relatively fragmented is that the assembly software retains more assembly possibilities, thus retaining a large number of short fragment sequences. This can be confirmed by the data from N50.
One suggestion for reference is to assemble strategy 1 and directly predict gene structure by removing sequences without genes. Subsequently, these genes should be functionally annotated, and based on the species information matched by the gene functional annotation on each contig, the target species should be retained or obvious non target species should be deleted. Usually, by this point, a relatively pure genome can be obtained. If not, strategy 3 can be executed based on this data.
The article used BUSCO to evaluate genome integrity, but missing duplicated genes, fragmentation, and missing genes are crucial for complex genome assembly and evaluation.
The article mentions QUAST in terms of methodology, but no relevant results have been seen.
The relationship between a few images and the main text is not significant. At least additional information should be provided, such as genome quality, repeat sequences, GC content of major contigs, structural integrity, gene density, etc,
L23, Mb should be Kb
L106-107, These abbreviations need to be explained in detail or provide complete formulas,
L119-122, The number of reads should be displayed in the results section. What needs to be displayed here is the sequencing data volume of the original design.
L223-225, To my knowledge, HiFi sequencing reads have high base quality and do not require polishing.
L278, I don't know the genetic distance between Lobaria pulsatiaria and the species studied in this article. In theory, better gene annotation results can only be obtained using species' advanced RNA data. Obtaining RNA of the target species is not difficult.
In addition, this article only used one Illiumia DNA sequencing and one HiFi sequencing, and the sequencing quantity is not high. If it were me, I would use ONT to obtain longer sequencing data, then combine 1-2 Illiumia DNA short fragment libraries to polish the ONT, and finally combine RNA data with other gene structure prediction strategies to obtain a more complete genome. Of course, potential non target contigs need to be excluded from this.
Author Response
Major comments
This study fails to deliver on its promise to identify "the most effective assembly strategy" for fungal genomes from metagenomic data. The evaluated strategies—hybrid metagenomic assembly (1), long-read assembly with short-read scaffolding (2), and hybrid assembly of filtered mycobiont reads (3)—yielded universally suboptimal results, with Strategy 2 (55.5 Mb, N50=142.8 kb, 519 scaffolds) declared "best" despite high fragmentation and failure to achieve chromosome-scale continuity. Crucially, Strategy 3 performed catastrophically (N50=16.6 kb, BUSCO=37.6%), yet the authors omitted root-cause analysis for this failure. The conclusions lack rigor due to unaddressed limitations: PacBio read lengths (avg. 5 kb) were insufficient for complex metagenomes, no benchmarking against state-of-the-art tools (e.g., metaFlye+Hi-C) was conducted, and claims of strategy superiority remain unsubstantiated beyond the authors' compromised dataset. The title overstates both methodological innovation and outcomes, as the work merely confirms known challenges without advancing scalable solutions.
ANSWER: We agree with the reviewer that the title may be confusing, and for this reason, it has been changed. Nevertheless, it is important to note that the primary objective of this study is not to generate a chromosome-level assembly. As the reviewer indicated, alternative strategies, such as Hi-C and Nanopore sequencing, could yield a superior assembly chromosome-level genome. Nevertheless, the aim of this study is to evaluate the performance of commonly used strategies when both Illumina and PacBio data are available, which are the most widely used technologies.
To avoid any potential confusion for readers, a paragraph has been added to the discussion. This paragraph indicates the methodological limitations of Illumina and PacBio, and the alternative approaches proposed by the reviewer to improving genome quality.
To improve the manuscript in line with the reviewer's general comments, the following changes have been made:
- The results of strategy 3 have been discussed in greater depth (lines 503-508).
- A paragraph has been added indicating alternative strategies that could have generated a genome at the chromosomal level (lines 544-549).
Detailed comments
PacBio reads averaged 5,012 bp—insufficient for resolving repetitive regions in complex metagenomes.
ANSWER: We agree that the average PacBio read length of 5,012 bp is a potential limitation for assembly chromosomal level assembly. In the discussion we have added a paragraph limiting our ability to resolve highly repetitive regions, especially due to the complexity of metagenomic data. We highlighted this limitation and how it could be contributing to the poor performance of our assembly strategies in the Discussion section. We also empathized the necessity of improving sequencing in future work.
In fact, I am more in favor of the outcome of Strategy 1, but the author gave up on it because it was too fragmented. The reason why Strategy 1 is relatively fragmented is that the assembly software retains more assembly possibilities, thus retaining a large number of short fragment sequences. This can be confirmed by the data from N50.
ANSWER: In order to verify that strategy 1 could generate good results, it was continued until the end, with the mycobiont contigs being filtered and the same scaffolding and polishing process as in strategy 2 being performed (Fig. 1, lines 139-156). The result of the final assembly of strategy 1 is still worse than that of strategy 2 (Table 2), in terms of number of scaffolds, N50, and BUSCO values.
The reviewer's explanation that retaining a large number of short contigs results in a more fragmented assembly has been added to the discussion. To Demonstrate that this is the cause, we show the total number of total contigs of the assembly, which is much higher than that of scaffolds > 500 bp.
One suggestion for reference is to assemble strategy 1 and directly predict gene structure by removing sequences without genes. Subsequently, these genes should be functionally annotated, and based on the species information matched by the gene functional annotation on each contig, the target species should be retained or obvious non target species should be deleted. Usually, by this point, a relatively pure genome can be obtained. If not, strategy 3 can be executed based on this data.
ANSWER: In order to compare the three strategies, we consider important to filter the mycobiont contigs using the same methodology across the strategies. Therefore, we employed BlobTools to filter the mycobiont contigs from strategy 1.
On the other hand, we preferred avoiding gene prediction prior to obtain the mycobiont genome for two reasons: (1) this process require a significant computational effort and a complete database and (2) the absence of well characterized proteomes of Peltigerales fungi could limit the efficiency and accuracy of gene prediction.
The article used BUSCO to evaluate genome integrity, but missing duplicated genes, fragmentation, and missing genes are crucial for complex genome assembly and evaluation.
ANSWER: We added the proportion of duplicated, fragmented and missing BUSCOs in tables 1 & 2.
The article mentions QUAST in terms of methodology, but no relevant results have been seen.
ANSWER: The metrics calculated using QUAST are no. scaffolds, length of largest contig, genome size, N50 and GC content. These data are included in table 1 & 2. To clarified it, in the captions of tables are mentioned that these valued were generated with QUAST.
The relationship between a few images and the main text is not significant. At least additional information should be provided, such as genome quality, repeat sequences, GC content of major contigs, structural integrity, gene density, etc,
ANSWER: Thank you for this suggestion. The gene density and GC content for each assembly have been added in the results. Additionally, as supplementary material, new figures representing the GC content of the largest scaffolds, the number of genes in groups of scaffolds grouped by size, and the distribution of gene density and scaffold size have been added. Information on genome quality and repeat content are in the tables.
L23, Mb should be Kb
ANSWER: This typo has been corrected.
L106-107, These abbreviations need to be explained in detail or provide complete formulas,
ANSWER: Done. The name of the technique, thin layer chromatography has been added. In addition, the composition of the solvents is also specified (Lines 107-110).
L119-122, The number of reads should be displayed in the results section. What needs to be displayed here is the sequencing data volume of the original design.
ANSWER: Done. The total number of reads was moved to results (lines 311-314) and the coverage design was added in methodology (lines 123-124).
L223-225, To my knowledge, HiFi sequencing reads have high base quality and do not require polishing.
ANSWER: Yes, it is true, and in fact we did not perform any quality filtering of HiFi reads. It is indicated in Quality Control and Read Filtering section (lines 130-131): “in the case of long reads, no quality filtering or error correction step was performed prior to assembly.”
The lines 223-225 referred to polish the mycobiont assembly. For genome polishing, we used the (uncorrected) long reads which were filtered to retain only those belong to mycobiont but no error correction or trimming was applied. We have clarified it in the text (lines 225-234).
L278, I don't know the genetic distance between Lobaria pulsatiaria and the species studied in this article. In theory, better gene annotation results can only be obtained using species' advanced RNA data. Obtaining RNA of the target species is not difficult.
ANSWER: Lobaria pulmonaria is closely related to S. crocea, both species belong to the same family, Peltigeraceae. The use of expressed sequence tags (ESTs) or transcriptomes from related species, even from different families, is a common approach in genome annotation. In fact, this approach has been successly used for the annotation of other lichenized fungi (Llewelyn et al., 2023a. Genome Biology and Evolution, 15(2), evad002; Llewelyn et al., 2023b. Genome Biology and Evolution, 15(5), evad074).
In methodology (lines 284-285) we have added the relationship between both species.
In addition, this article only used one Illiumia DNA sequencing and one HiFi sequencing, and the sequencing quantity is not high. If it were me, I would use ONT to obtain longer sequencing data, then combine 1-2 Illiumia DNA short fragment libraries to polish the ONT, and finally combine RNA data with other gene structure prediction strategies to obtain a more complete genome. Of course, potential non target contigs need to be excluded from this.
ANSWER: The sequencing strategy proposed by the referee is appropriate for obtaining a genome at the chromosomal level. However, this is not the objective of the study. In any case, we have added a paragraph in the discussion section indicating that this would be the ideal strategy for obtaining a higher quality genome.
Round 2
Reviewer 2 Report
None
The author's response has been fully addressed my concerns. The quality of the paper has apparently improved. I agree with the publication of this article.
Reviewer 3 Report
I have no further comments
I have no further comments